# Adipose Stem Cells in Modern-Day Ophthalmology

**Mutali Musa** [1] , **Marco Zeppieri** [2,*] , **Ehimare S. Enaholo** [3], **Carlo Salati** [2] **and Pier Camillo Parodi** [4]

1  Department of Optometry, University of Benin, Benin City 300238, Nigeria
2  Department of Ophthalmology, University Hospital of Udine, 33100 Udine, Italy
3  Centre for Sight Africa, Nkpor, Onitsha 434112, Nigeria
4  Department of Plastic Surgery, University Hospital of Udine, 33100 Udine, Italy
*  Correspondence: markzeppieri@hotmail.com

**Abstract:** Stem cells (SCs) have evolved as an interesting and viable factor in ophthalmologic patient care in the past decades. SCs have been classified as either embryonic, mesenchymal, tissue-specific, or induced pluripotent cells. Multiple novel management techniques and clinical trials have been established to date. While available publications are predominantly animal-model-based, significant material is derived from human studies and case-selected scenarios. This possibility of explanting cells from viable tissue to regenerate/repair damaged tissue points to an exciting future of therapeutic options in all fields of medicine, and ophthalmology is surely not left out. Adipose tissue obtained from lipo-aspirates has been shown to produce mesenchymal SCs that are potentially useful in different body parts, including the oculo-visual system. An overview of the anatomy, physiology, and extraction process for adipose-tissue-derived stem cells (ADSC) is important for better understanding the potential therapeutic benefits. This review examines published data on ADSCs in immune-modulatory, therapeutic, and regenerative treatments. We also look at the future of ADSC applications for ophthalmic patient care. The adverse effects of this relatively novel therapy are also discussed.

**Keywords:** stem cells; adipose; pluripotent; regenerative

## 1. The Anatomy of Adipose-Tissue-Derived Stem Cells (ADSC)

Adipose-tissue-derived stem cells (ADSCs) are mesenchymal cells derived from adipocytes, as the name implies [1]. The proliferative actions of these cells were discovered as secondary effects of leptin secretion [2–4]. Adipose tissue is distributed normally around the body at sites such as the bone marrow (i.e., red adipose) [5], articular cartilage, and visceral and subcutaneous fat deposits [6–10]. Adipose tissue may also form around the hepatic tissue [11] and cardiac smooth muscle [12]. Adipose tissue obtained via lipo-aspirates can be analyzed in an alkaline medium catalyzed by collagenase enzymes [13]. The subsequent reactions help reveal the base anatomy. Adipose tissue is made up of a stromal vascular fraction (SVF) and mature adipocytes [14].

Various cellular components make up the SVF [15]. These include adipose stromal/stem cells, pericytes, preadipocytes, smooth myocytes, myeloid cells, fibroblasts, endothelial cells, lymphocytes, and macrophages [16]. Of these constituents, adipose stromal cells are identified as being experimentally multipotent [17]. It is important to remember that adult mesenchymal SCs are derived from the stromal vascular fraction component of adipose tissue [18].

Adipose tissue exists in various isoforms that chiefly differ in their thermogenic capacities. These include brown adipose tissue [19] and white adipose tissue [20]. White adipose tissue exists at subcutaneous sites, in the bone marrow, and around the linings of visceral organs [21]. Brown adipose tissue, however, is located all around the body and helps to convert energy from food into heat, especially in cold temperatures [22]. Brown adipose tissue also contains more mitochondria and capillaries than white adipose tissue, indicating that it is more efficient at supplying oxygen to surrounding tissues [23]. Cadherins, along with integrins, are known receptors that mediate transmembrane adhesion [24].

## 2. A Brief Introduction to Stem Cells (SCs)

SCs can be seen as undifferentiated, unspecialized cells possessing characteristics that allow them to proliferate and integrate into differentiated cells under prime physiological conditions [24]. The vast potential of SCs for various aspects of tissue regeneration, remodeling, and reconstitution has been studied over the past several years, with the earliest therapeutic applications emerging in the fields of oncology and bone marrow transplantation [25,26]. SCs can be considered either embryonic stem cells (ESCs) or somatic stem cells (SSCs) [27], which vary in their differentiation capabilities depending on the stage of development. In general, stem cells can be classified as pluripotent [28], totipotent [29], oligopotent, multipotent, or unipotent [30]. ESCs tend to possess a much greater disposition for pluripotency than somatic stem cells [31]. Research on human embryonic stem cells (hESCs) has targeted ways to catalyze cellular pluripotency.

Totipotent stem cells (TSCs) can differentiate into all tissues present in the entire living organism, which includes both embryonic and extraembryonic constituents [32]. Embryologically, upon the fertilization of the ova, a zygote is formed with a high degree of totipotency. At this stage, it is programmed to differentiate into multiple germ layers and the placenta. After about a week, the embryo travels down the fallopian tube and differentiates into the blastocyst, which is formed by a blastocoel, inner cell mass, and trophectoderm. The blastocyst inner cell mass consists of pluripotent stem cells (PSCs).

PSCs can differentiate into the three germ layers but cannot proliferate to extraembryonic elements (placenta) [33], which is possible with totipotent cells. Pluripotent cells exist as ESCs within the blastocyst of the embryo during the early stages of fertilization. Based on in vitro studies, it is known that PSCs are derived from the inner cell mass of pre-implantation embryos [34]. Over the years, the ethics of working and researching with these cells has been questioned [35,36]. The discovery of induced pluripotent stem cells (iPSCs), however, via reversal from somatic stem cells, has aided in addressing these ethical barriers. ESCs are termed iPSCs when hESCs are derived from the epiblast layer in the post-implantation stage [37,38].

From PSCs, multipotent, oligopotent, and unipotent SCs emerge down the differentiation lineage. Multipotent SCs possess characteristics to differentiate into cells originating from specific lineages [39]. Hematopoietic SCs represent a common lineage of multipotency [40]. Oligopotent cells, in comparison, have a narrower spectrum of differentiation into several (very related) cell types [40,41]. Unipotent cells, however, have the most limited differentiation capacity but possess special properties for repeated cellular division [42].

Adult or somatic stem cells (SSCs) exist as unspecialized, undifferentiated cells in the midst of other differentiated cells [43]. These cells can be found in various tissues upon the completion of the developmental process. SSCs play advanced roles in tissue repair, replacement, and wound healing when placed in the right physiological environment [44]. Most somatic SCs have a limited capacity for differentiation. Out of a broad range of adult SCs, mesenchymal stem cells (MSCs) are the most significant [45]; others include neural cells, hematopoietic SCs, etc.

MSCs differentiate into several tissues of a multicellular lineage. These cells possess broader multipotent characteristics (via iPSCs) when compared to other types of SSCs [46]. MSCs have been reported to play various roles in biological tissue formation and regeneration. MSCs are important in the repair and regeneration of bone, cartilage, adipose tissue, bone marrow, tendons, and muscle. MSCs can thus be seen as progenitors for differentiating into adipocytes, chondrocytes, myocytes, and osteocytes [47–49]. Reversibly, they can also be derived via these aforementioned cell lineages [50]. The occurrence of these processes depends on trans-differentiation.

### 3. Introduction to Stem Cell Medicine

Stem cell/regenerative medicine has been developed from the discovery that SCs can be modulated accordingly for the potential enhancement of variable tissue repair and regeneration [51]. Numerous strategies have been proposed to achieve programmed SC differentiation, arising from pertinent research, experimental models, and expansive future potential. Tissue engineering, molecular/cellular biology, and medical science make up the basis of SC medicine [52]. The discovery of adult MSCs has paved the way for game-changing possibilities in the field of SC medicine. MSCs are progenitors for the differentiation of cells of mesenchymal origin (i.e., adipocytes, chondrocytes, myocytes, and osteocytes). These cells have been primarily studied, and numerous biological applications have been hypothesized to date.

In stem cell research, MSCs are commonly derived from autologous, allogeneic, and xenogeneic sources [53]. Experimentally, they are often isolated from the bone marrow; however, gaining access to bone-marrow-derived MSCs requires significantly invasive aspiration techniques [54]. Upon the onset of senescence, however, MSCs derived from the bone marrow often exhibit depreciated proliferative capabilities [55,56]. Adipose tissues, in comparison, are a preferred source of adult MSCs. Adipose tissue can be harvested using minimally invasive techniques with far less associated risk than bone marrow aspiration [57]. These cells tend not to lose the capacity to differentiate upon the onset of senescence [58]. Hence, adipose tissue represents a more viable "lifelong" source of MSC.

The effects of adipose-derived MSCs (ADSCs) on multiple acquired and degenerative disorders have been investigated in numerous animal-based research studies but sparingly in human trials. Theoretically, ADSCs have been found to have therapeutic significance around the site of delivery via immunosuppressive [59], angiogenic [59], and proliferative mechanisms [60]. ADSCs also promote tissue regeneration via the paracrine secretion of extracellular vesicles, which have been observed to release trophic factors in vivo [61]. Known paracrine factors secreted by adipose stem cells include matrix metalloproteinase inhibitors (−1, 2, and 3), vascular endothelial growth factor [62,63], platelet-derived growth factor [64], tumor necrosis factor-alpha [64], human growth factor, various cytokines, hepatocyte growth factor [65], transforming growth factor beta-1 [66], etc.

The extracellular vesicles of adipose tissue have been hypothetically viewed as secretomes for the potential development of "cell-free" regenerative therapies. The future discovery of alternative cell-free strategies could help ease ethical concerns limiting the unrestricted practice of SC medicine in modern times.

### 4. Extraction of Adipose-Tissue-Derived Stem Cells (ADSCs)

Adipose tissue can be extracted during elective surgery with little to no risk to the donor. It is also possible to extract adipose tissue using a fine-needle aspiration technique. The extracted adipose tissue is easily cryopreserved for extended periods with minimal loss in viability [67]. ADSCs are then isolated from adipose tissue using collagenase digestion. Isolated SCs display differing characteristics depending on the site of extraction. Studies have shown the possibility of extracting different progenitor cells from ADSCs obtained during eyelid surgery [68]. The authors described a novel method of extracting these cells by digesting ADSCs using collagenase for 16 h. The resulting cells were expanded and exhibited an ability to differentiate into endothelial cells and osteocytes, amongst others. They went on to hypothesize that central and medial orbital adipose tissues could be a source of both mesodermal and neuroectodermal progenitor cells, respectively.

Further studies have assessed the viability of adipose stem cells derived from fat harvested via either tissue resection or power-assisted liposuction [69]. Cell viability was determined using fluorescence-activated cell sorting and amine-reactive dyes. The researchers subsequently found that cell viabilities and senescence ratios were similar for both harvesting techniques. The power-assisted liposuction technique was reportedly preferred for its reduced invasiveness.

## 5. Methods Used in Minireview

### 5.1. Inclusion Criteria

- Records that focused on the use of adipose stem cell tissue in ophthalmology;
- Studies on the use of stem cells in human tissue;
- Full-text papers;
- Journal articles published within the last 10 years, ranging from 2012 to 2022;
- Studies published in the English language;
- Primary sources with qualitative or quantitative research designs.

### 5.2. Exclusion Criteria

- Animal and animal-tissue-based studies;
- Studies that focused on other fields of medicine outside ophthalmology;
- Studies not published in English;
- Journal articles published outside the range from 2012 to 2022

To ensure the accuracy and transparency of the records used for this study, the Preferred Reporting Items for Systematic Reviews and Meta-Analyses (PRISMA) [70] was used to populate this review in October 2022 using the PubMed database, which is shown in the chart below in Figure 1.

In the PRISMA chart, i is the total number of records identified through the PubMed search (150); ii is the net number of articles identified after removing duplicates (147); iii is the number of articles that passed the screening criteria to be included in this review (147); iv is the number of records that did not meet the screening criteria (37); v is the number of full-text articles among the records that met the screening criteria (110); vi is the number of articles further excluded with reasons (41); and vii is the final number of articles used in this review (69).

The following describes the PubMed search criteria for ADSCs in ophthalmology. The inclusion criteria limited articles to publications involving the use of adipose-tissue-derived stem cells in an ophthalmology setting. The earliest year of publication of the articles was set to be a maximum of 10 years before writing this paper (2012). Only articles that were available for free and as full texts were included. One article was removed because it was published in a language the authors could not read.

PubMed keywords were "Adipose-derived" AND "stem cells" OR "stem" AND "cells" OR "stem cells" AND "ophthalmologie" OR "ophthalmology" OR "ophthalmology" OR "ophthalmology".

The search strategy was "Adipose-derived"[All Fields] AND ("stem cells"[MeSH Terms] OR ("stem"[All Fields] AND "cells"[All Fields]) OR "stem cells"[All Fields]) AND ("ophthalmologie"[All Fields] OR "ophthalmology"[MeSH Terms] OR "ophthalmology"[All Fields] OR "ophthalmology s"[All Fields]).

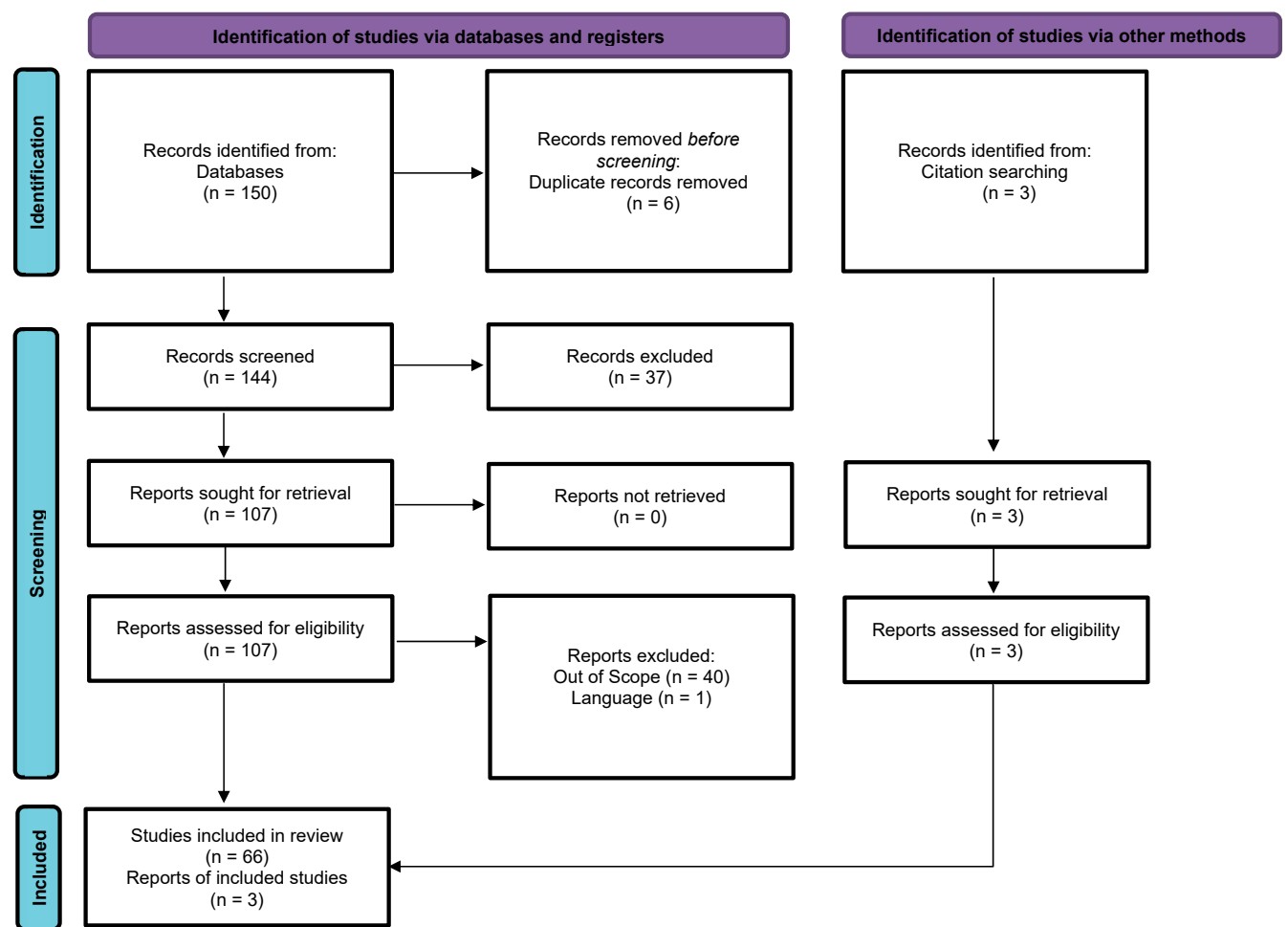

**Figure 1.** PRISMA flow diagram of included studies, adapted from Ref. [70].

## 6. Delivery of Autologous or Allogeneic Adipose-Derived Stem Cells for Experimental Management of Ocular Surface Disease (OSD)

A review of specific uses of ADSCs in ophthalmology suggests that the ocular structure of interest is a determinant for the optimal selection of delivery protocols. Numerous studies in the literature in this field have shown the possibilities of ADSC therapy to treat various types of ocular surface diseases (OSDs), including chemical burns, corneal epitheliopathy, corneal laceration, dry eye disease, limbal stem cell deficiency, keratoconus, etc. [71].

The cornea is a trilaminar structure. The epithelial layer is derived from the surface ectoderm; the stroma is from neural crest mesenchyme; and the endothelial layer is from neural crest cells. Considering the multicellular differentiation potential of corneal cells, the adoption of ADSCs in corneal treatment could hypothetically improve the availability of viable lamellar tissue [72]. ADSC-conditioned mediums have been shown to enhance the viability of corneal endothelial cells [73]. Alternative strategies, such as transplanting human ADSCs (hADSCs) on synthetic hyaluronic acid scaffolds into the corneal stroma of murine eyes via created flap incisions, have shown potential for the differentiation of ADSCs into a collagen matrix [74]. ADSC exosomes have also been reported to promote corneal stromal fibroblast viability [75]. Lee et al. tested a topical application of low-molecular-weight ADSC-conditioned medium (LADSC-CM) on dry eye syndrome murine models and concluded that ADSCs were useful in alleviating signs of dry eye [76].

Espandar et al. grafted xenogeneic hADSCs onto acutely afflicted rabbit corneas using a scleral contact lens carrier. The study showed a reduction in ocular inflammatory signs that were not observed in controls [77]. A similar study on in vivo rabbit models via

subconjunctival injection of an hADSC suspension showed favorable results in chemical burn recovery and corneal tissue repair that were not manifested in the controls [78]. Experiments performed by our group a few years ago utilized the topical delivery of stem cells derived from processed human lipo-aspirates in rat models with alkaline corneal injury [79]. Our study showed complete and enhanced healing times for the re-epithelialization of lesioned rat corneas in the ADSC-treated group when compared with the other treatment and control groups. Histological samples showed fewer inflammatory markers and potentially better healing mechanisms in the ADSC group.

Corneal neovascularization readily occurs when there is an insult to the corneal integrity [80]. Epithelial–mesenchymal transition (EMT) has been implicated in promoting subepithelial corneal fibrosis in limbal stem cell deficiency. The secretome of ADSCs has demonstrated potency in inhibiting the EMT of corneal epithelial cells, thereby reducing corneal tissue fibrosis [81]. Pirounides et al. delivered autologous ADSCs to rabbit corneal lesions via topical, subconjunctival, and intrastromal routes [82]. ADSC-treated corneal tissue showed statistically reduced corneal neovascularization when compared to a control group treated with phosphate-buffered saline.

### 7. ADSCs in Ophthalmic Surgery

Ophthalmic surgery is often necessary for the management of ocular morbidity. Surgery can offer therapeutic benefits; however, ocular complications such as trauma must be considered due to specific procedures, inflammation, and infection. Penetrating keratoplasty and keratoprosthesis are invasive treatment options for visual impairment due to extensive corneal scarring. The use of donor corneas has numerous limitations, which include a lack of donor tissue, the possible rejection of donor tissue, high complication rates, elevated risks of postoperative complications, increased induced astigmatism, etc. [74].

Less invasive options are becoming of increasing interest in the field regarding corneal repair. Studies have shown that exosomal microRNA-19 suppresses the differentiation of corneal stromal keratocytes into myofibroblasts [83]. Ma et al. grafted autologous rabbit adipose stem cells on a poly lactic-co-glycolic acid (PLGA) scaffold following mechanical corneal stromal injury. Marked stromal repair was reportedly observed in vivo [84]. Our group tested the efficacy of topical ADSCs on mouse models with laser-induced photorefractive keratectomy [85]. Homogeneous corneal lesions were created using a laser device. Fluorescein staining and specific corneal photography were used to estimate the extent of the lesions during the entire treatment regimen. The study showed that the group of mice that received supplemental ADSC topical treatment healed faster and with fewer complications when compared to the other groups and the controls.

ADSC therapy has been shown to enhance surgical outcomes and reduce postoperative complications. El Zarif et al. reviewed the laser-assisted intrastromal implantation (via injection) of autologous adipose-derived stem cells in patients with advanced keratoconus to aid in corneal stroma regeneration [71]. They reported an improvement in corneal integrity and function in all of their test groups, which received some form of stem cell therapy. Similar exciting results were reported by Qiu et al., who performed sclerocorneal transplantation of amniotic membranes and autologous ADSCs onto the ocular surfaces of rabbits with limbal stem cell deficiency [86]. ADSC therapy alone performed better than a combined ADSC and amniotic membrane therapy or a third placebo group.

Park et al. utilized a topical cell-free conditioned medium derived from autologous ADSCs on murine models of alcohol-burn-induced keratitis [87]. The ADSC preparation was administered four times daily in the test mice, while other groups of mice received a placebo. Fluorescent biomicroscopy showed the improved expression of corneal epithelial cells. There was also an upregulation of interleukin-6, epidermal growth factor, and C-X-C chemokine receptor type 4 mRNAs. Shadmani et al. injected both autologous and allogeneic ADSCs subconjunctivally in animal models of corneal alkali injury and obtained similar results [88]. Dinç et al. also performed the subconjunctival injection of

ADSCs in murine models of acute alkaline corneal burns and demonstrated increased wound healing [89].

Bone-marrow-derived mesenchymal tissue and ADSCs have been compared in several studies in the literature. Demirayak et al. created experimental models to study the mitigating effects of SCs on corneal scarring following penetrating injuries [90]. They performed intracameral injections of allogeneic ADSCs in Wistar rats following induced penetrating corneal injury. They concluded that allogeneic treatment using SCs precipitated the regeneration of damaged cornea stroma and a reduction in subsequent scarring. Shang et al. also assessed the impact of allogeneic ADSCs on the healing response of murine corneas following ethanol exposure via retrobulbar injection and reported that ADSCs were found to promote the clearance of neutrophils in the cornea during the granulation stage [91]. This was highlighted as a key step in a cascade of events that reduced the amount of corneal scarring in their model.

## 8. Therapeutic Applications of ADSCs in Retinal Diseases

ADSCs have been the topic of several studies in the literature that examine the potential therapeutic options for these cells in retinal diseases. Rajashekar et al. found that intravitreal injection of ADSCs in streptozotocin-induced diabetic rats correlated with fewer signs of early vascular derangement characteristic of diabetic retinopathy (DR) [77]. Safwat et al. also reported that adipose-stem-cell-derived exosomes ameliorated characteristic retinal degeneration following intraocular and subconjunctival administration among streptozotocin-induced diabetic rabbits [92].

As with any surgical procedure, ADSC implantation has been assessed for potential risks to the retina. Limoli et al. recorded no complications and an improvement in scotopic electroretinographic scores following suprachoroidal grafting of mature adipocytes and ADSCs in the stromal vascular fraction enriched with platelet-rich plasma (PRP) amongst elderly patients with non-exudative age-related macular degeneration (AMD) [93]. This group also reported objective parameters following the autologous transplantation of adipocytes, ADSCs in the SVF, and platelet-rich plasma (PRP) within the suprachoroidal space of patients with dry AMD [94]. The improved visual performance also correlated with greater retinal thickness averages. Limoli et al. reported that the suprachoroidal transplantation of autologous ADSCs reduced the progression of visual deficits in dry AMD, as shown by an improvement in best-corrected visual acuity (BCVA) and logMAR values 180 days post-treatment. A longitudinal study of visual characteristics from eight patients showed a slight improvement in the visual function of patients with degenerative macula disease following suprachoroidal ADSC implantation [95].

## 9. Future Potential for ADSCs in Retinal Diseases

Genetic and tissue engineering has been hypothesized for the treatment of DR [96]. Vision loss from advanced ischemic or late (proliferative) DR is often irreversible. Therapeutic advantages are best seen with treatment during the earlier stages of DR [97]. Modern-day strategies mostly recommend hyperglycemic control (as well as the control of other vasculopathic patient-specific risk factors) during the mild and early moderate stages of non-proliferative DR. Improving glycemic control following prolonged periods of retinal exposure to hyperglycemic stress, however, can still cause non-reversible effects of previous retinal microvascular damage. Laser or intravitreal treatments are usually performed when there are signs of active DR. ADSCs have been suggested to provide protective effects against retinal ischemic damage [98] and to retinal extracellular vesicles [99].

Rajashekar et al. proposed a strategy to regenerate the retinal vasculature and neuronal cell integrity via intravitreal injection of ADSCs in streptozotocin-induced murine models of early DR [100]. ADSCs were reported to integrate into retinal perivasculature and, thus, reconstitute the blood–retinal barrier within several weeks.

It has also been postulated that the administration of autologous ADSC exosomes to streptozotocin-induced diabetic rabbits could attenuate diabetic-retinopathy-related

neurodegeneration and microvasculopathy via increased microRNA-222 expression [92]. Elschaer et al. subsequently concluded that intravitreal injections of either hADSC- or adipose-SC-conditioned medium primed with cytokines yielded a reduction in vascular permeability and an improvement in electroretinogram scores [101]. ADSCs have also been found to mediate angiogenesis via paracrine mechanisms in retinal endothelial cells and promote retinal regeneration in vitro [102]. ADSC-CM and paracrine factors were associated with better visual functions post-injection in early DR Ins2Akita mouse models.

ADSCs are known to play a role in retinal and photoreceptor cell proliferation [103]. hADSCs have shown a trilineage potential to proliferate, migrate, differentiate into RPE cells when exposed to an RPE-cell-conditioned medium [104]. Yu et al. reported that adipose-derived mesenchymal stem cell exosomes ameliorated laser-induced retinal injury among mice and prevented extensive photoreceptor cell damage via the downregulation of monocyte chemotactic protein-1 [105]. Xu et al. also found that orbital ADSCs isolated via direct explant culture showed the earlier and stronger expression of markers indicating eye field and retinal photoreceptor differentiation than those generated by the conventional enzyme method [106].

## 10. Therapeutic Potential of ADSCs in Neuro-Ophthalmology

Late-phase optic nerve disease yields profound functional limitations, which usually tend to be irreversible [107]. ADSCs, however, may produce a paradigm shift if viable therapies can be established. ADSCs have been reported to offer neuroprotection to retinal ganglion cells and may promote the regeneration of axons in the optic nerve head via the secretion of trophic "paracrine" factors [108]. ADSC exosomes possess bioactive molecules such as microRNAs and immunoregulatory, trophic, and growth factors, which provide pro-angiogenic effects for the possible re-vascularization of ischemic retinal or neural tissue [109]. Faber et al. suggested that ADSCs could be administered intravitreally without adverse consequences [110]. The results were based on clinical findings in a single patient with non-arteritic anterior ischemic optic neuropathy (NAION). Oner et al. reported a slight improvement in visual acuity among patients with optic atrophy following the suprachoroidal implantation of ADSCs. This patient also showed visual field and mfERG recording improvements after treatment [111].

Experiments based on in vivo ADSC transplantation have shown possible therapeutic benefits in rat models with ameliorated optic nerve injury. Treatment showed signs of inhibiting insult-induced inflammation [112,113]. The effects of ADSCs on optic nerve injuries have also been studied in Sprague Dawley rats. Experiments showed that an adipose SC concentrated conditioned medium (ASC-CCM) primed with inflammatory cytokines to induce the expression of tumor necrosis factor-stimulated protein 6 (TSG-6) improved the retinal barrier function and reduced visual deficits via neuroglial support mechanisms when injected intravitreally following mild traumatic brain injury in mice [114,115]. Jha et al. also highlighted the potential of the ADSC concentrated conditioned medium for regulating retinal neurodegeneration following mild traumatic brain injury [116].

## 11. Potential Applications of Adipose Stem Cells to Periorbital and Orbital Structures

Studies have reported that ADSC therapy could be beneficial in thyroid-associated orbitopathy [117]. It has also been suggested that insulin-like growth factor 1 has pro-adipogenic and pro-proliferative effects on ADSCs extracted from thyroid-associated ophthalmopathy patients via in vitro studies [118]. The topical prostaglandin analog bimatoprost has been reported, however, to inhibit human orbital ADSCs [119].

Wu et al. reviewed the potential of iPSCs derived from autologous adipose tissue to forge patient-specific remedies for degenerative and acquired oculoplastic diseases [120]. Lee et al. reported orbital volume augmentation following the intraorbital injection of ADSCs in rabbits [121]. The rabbits showed an increase in the exophthalmometric value of about 2.5 mm after three months. Li et al. induced autoimmune dacryoadenitis in rabbits via the intravenous infusion of activated autologous peripheral blood lymphocytes. Acute

treatment using ADSCs was attributed to the reduced expression of inflammatory markers and increased basal tear volume [122].

## 12. Complications and Limitations with Intraocular ADSC Transplantation

Negative outcomes have been reported by Fuentes-Julián et al., who sought methods to reduce the rejection of corneal grafts [123]. They injected ADSCs into the stroma at the allograft junction in "normal-risk" transplanted rabbit corneas. The systemic intravenous administration of ADMSCs was utilized in animals with "high-risk" allograft rejection. The study showed exacerbated inflammation and shorter graft retention rates in treated animals.

Oner et al. administered subretinal ADSCs in a small sample of patients with advanced retinitis pigmentosa. Post-implantation reports indicated that 54% of the study group developed complications around the site of implantation, and 90% of the study group reportedly showed no objective evidence of improved visual function post-therapy [124]. Kuriyan et al. also reported the loss of vision in three patients with AMD following the intravitreal administration of autologous ADSCs [125].

Deleterious effects of in vivo exogenous mesenchymal stem cell implantation have been reported to occur via myofibroblast proliferation around microvascular smooth muscle cells and pericytes [126]. Findings of vitreous hemorrhage and bilateral tractional retinal detachment were reported in a 77-year-old female with exudative macular degeneration three weeks post-intravitreal injection of ADSCs [127].

## 13. A Look into the Future of Stem Cells in Ocular Surgery and Management

Cataract is one of the leading causes of avoidable visual impairment. ADSCs have the potential to offer future management strategies. ADSC exosomes have been found to attenuate UV-induced lenticular changes via the downregulation of cartilage acid protein-1, CRTAC1 [128]. Zhou et al. demonstrated that autologous or allogeneic ADSCs could integrate into trabecular meshwork (TM) cells when delivered into the anterior chambers of murine models [129]. ADSCs could also reportedly undergo targeted differentiation into viable TM cells in vitro. Further studies are required to understand the potential effects of aqueous humor and in vivo myocilin proteins to induce the osteogenesis of ADSCs. This could be of utmost importance to prevent the possible calcification of anterior chamber structures post-intracameral injection [130]. Further research is also needed to establish the exact mechanisms of optic neuroprotection and other postulated mechanisms of facial nerve regeneration via MSCs [131].

The further development of strategies to advance viable neuroectodermal progenitor cell extraction from orbital adipose tissue could broaden the range of differentiation of ADSCs, especially regarding their integration into non-mesenchymal ocular structures. Future studies could provide hope for the generation of viable tissues for corneal transplant without rejection. Studies unfortunately have yet to isolate ADSCs from specific orbital adipose tissue differentiated from the neural crest, from which corneal tissue also embryologically derives.

Cytokine-primed MSCs expressing the anti-inflammatory protein TSG-6 could potentially find future applications in the clinical picture of idiopathic orbital inflammatory syndrome. Cytokine priming has also been suggested to have a role in retinal vasculopathies via cytokine-primed adipose stem cells' protection against increased vascular endothelial permeability associated with DR [132]. Exosomes of ADSCs also exhibit potential benefits for the reversal of acute-phase ischemic retinal disease [109].

Espandar et al. suggested that hADSCs possess the capacity to differentiate into keratocytes within the corneal stroma in animal models [74]. This holds promise for the future development of alternative strategies in the management of visual impairment due to significant corneal stromal opacification. The directed differentiation of hADSCs to corneal endothelial cells may also improve the availability of replacement tissue [133,134]. ADSCs may also be useful in treating and managing other diseases and corneal conditions and refraction [135–139].

Limbal stem cell (LSC) insufficiency can present with severe visual deficits, including light sensitivity and pain [140]. Human feeder ADSCs have been successfully shown to support the growth of LSCs in vitro in cell clusters as compared to using single cells [141]. Oliva et al. reported a clinical trial of 249 patients with limbal stem cell deficiency who were managed with oral-mucosa-derived MSCs, with an improvement seen in 52.8% of the transplant-receiving corneas [142].

There is also promise in the use of ADSCs in managing individuals suffering from dry eye disease. A study on canines showed that injecting ADSCs into the lacrimal glands resulted in a significant increase in tear volume as measured by Schirmer's test [143]. Alió Del Barrio et al. reviewed human studies on the effect of ADSCs on corneal regeneration and adapted the methods therein for their research [77]. They were able to achieve ADSC differentiation into corneal endothelial cells.

## 14. Conclusions

In addition to their abundance, the minimally invasive approach for obtaining ADSCs renders these cells an interesting and viable option in various treatment regimens in ophthalmology. The multicellular lineage of ADSCs also provides the required diversity for ophthalmic applications. The adoption of ADSC therapies for refractory inflammatory and degenerative eye diseases holds great promise, considering the extensive differentiation capacities and secreted paracrine factors of ADSCs. Several studies in the literature, however, have reported the potential for unregulated cell proliferation when using these cells; thus, future long-term results are needed to explore the possible neoplastic effects of these cells when used for treatment. Caution is of utmost importance when considering cell-based therapies in human subjects, especially when treatment involves invasive and/or posterior segment delivery.

**Author Contributions:** Conceptualization, M.M. and M.Z.; methodology, M.M., M.Z. and E.S.E.; validation, M.M., M.Z., C.S. and P.C.P.; formal analysis, M.M., M.Z. and E.S.E.; investigation, M.M. and E.S.E.; resources, C.S. and P.C.P.; writing—original draft preparation, M.M. and M.Z.; writing—review and editing, M.M., E.S.E. and M.Z.; visualization, M.M., M.Z., E.S.E., C.S. and P.C.P.; supervision, C.S. and P.C.P.; project administration, C.S. and P.C.P. All authors have read and agreed to the published version of the manuscript.

**Funding:** This research received no external funding.

**Institutional Review Board Statement:** Not applicable.

**Informed Consent Statement:** Not applicable.

**Data Availability Statement:** Not applicable.

**Conflicts of Interest:** The authors declare no conflict of interest.

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
