# Peer review of "Adipose Stem Cells in Modern-Day Ophthalmology"

_clinpract, doi:10.3390/clinpract13010021_

Round 1
Reviewer 1 Report
The authors presented a well written introductory review of the potential for adipose-derived stem cells to treat various ophthalmic diseases. The strength of the the paper is its broad scope, but this is its key weakness as well. The PRISMA analysis lends transparency but also reveals that a significant proportion of the papers identified in a Pubmed search were excluded for 'reasons', in many cases simply because the papers were not available in full text format for free. I would prefer scientific reasons for inclusion and exclusion. But in an introductory review this is acceptable. Although I'm not sure the PRISMA approach added much to the analysis. The flowchart should be revised prior to publication to give it a more professional appearance.
The paper has an overall positive outlook on the potential for adipose-derived stem cells. This initially concerned me, because there are several one-liners about the potential to treat numerous corneal, retinal, neurologic, and oculoplastic diseases with undifferentiated stem cells. I am most familiar with the retinal literature, in which there have been significant examples of harm to patients with this approach. The authors address this with a section on risks associated with ADSC treatments, and the conclusions are cautionary. Therefore, I found the authors' approach acceptable.
Reviewer 2 Report
The authors give us a literature review regarding applications of adipose tissue-derived stem cells. The most papers in this field describe animal studies. Still little is known about possible clinical applications and their consequences in humans.
Can you provide any information about the use/clinical trials in humans?
Are there any trials ongoing?
Limbus stem cell insufficiency is a serious challange for many ophthalmologists.
Can the ADSC theoretically (or mayby experimentaly) differentiate into limbus stem cells?
How would the application look like?
How does it look with endothelial cells?
Are there any trials looking for differentiation of ADSC into corneal enothelial cells? If yes, how far are they?
Line 228: Nowadays, keratoplasties (which is a lamellar procedure in at least half of the cases) are not operations with high risk of postoperative complications.
Maybe you can reformulate that sentence.
